# Association of Depression and Excessive Daytime Sleepiness among Sleep-Deprived College Freshmen in Northern Taiwan

**DOI:** 10.3390/ijerph16173148

**Published:** 2019-08-29

**Authors:** Meng-Ting Tsou, Betty Chia-Chen Chang

**Affiliations:** The Department of Family Medicine, Mackay Memorial Hospital, Taipei City 10449, Taiwan

**Keywords:** college freshmen, excessive daytime sleepiness (EDS), depression

## Abstract

Background. The aim of this study was to investigate depression and other determinants (sleep-deprived behaviors such as hours spent sleeping, watching television, and on the computer) and their association with excessive daytime sleepiness (EDS) among college freshmen. Methods. Self-administered questionnaires were collected from two colleges in northern Taiwan from July to September 2014. A total of 2643 students (38.7% male; ages ranged 18–23 years; mean age of 18.8 ± 1.2 years) completed an anonymous questionnaire on lifestyle behaviors (including personal habits, sleep duration and quality, and hours spent watching television and on the computer); perception of one’s health, a validated depression scale (Brief Symptom Rating Scale, BSRS-5); insomnia symptoms (the Chinese version of the Athens Insomnia Scale, CAIS); and EDS rated with the Chinese Epworth sleepiness scale (CESS). The data were analyzed using the chi-squared test, *t*-test, multivariate logistic regression, and multiple linear regression. Results. The prevalence of EDS among college students was approximately 27.1% (717/2643). The risk of EDS was elevated with increasing severity of depression: odds ratio (OR) = 2.8/3.71/5.01 for female, and OR = 3.29/5.07/5.07 for mild/moderate/severe depression for male, respectively (*p* < 0.05; marginally higher in male severe depression, *p* = 0.08). If depression score increased by 1 point, CESS score increased by 0.35 point; if time spent on the computer during non-holidays increased by 1 h, CESS score increased by 0.1 point; and for those whose sleep duration increased by 1 h during non-holidays, CESS score decreased by 0.1 point. Conclusions. EDS significantly predicted depression among college freshmen. Using a computer for a long time and less sleep duration during non-holidays contributed to EDS of college freshmen. Youths who experience EDS are recommended to seek assessment for depression symptoms and sleep-deprived behaviors, thus allowing physicians to offer appropriate screening and treatment.

## 1. Introduction

Excessive daytime sleepiness (EDS) is a sign of inadequate sleep, whether due to sleep restriction/deprivation, or due to medical or psychiatric illness. EDS has been associated with risk of drowsy driving [1,2,3,4], injuries in the workplace [5,6], and poor school performance [7,8,9]. It has become a major international health concern [8,9,10]. The prevalence in the general population was reported to be about 3–13% in the US, 10% in Europe, 9% in Singapore, and 7% in Taiwan according to prior studies [11,12,13,14]. The prevalence of adolescent insomnia, EDS, and sleep insufficiency was 14–33% [15]. One study on 1056 young Chinese high school seniors (aged between 13 and 16) suggested a prevalence of 17.9% in that population [16]. Few studies have focused on college students in Asia; thus, it is important to investigate the risk factors behind this threatening problem and find ways to prevent it.

In 1990, the International Classification of Sleep Disorder (ICSD) divided EDS into four levels (normal, mild, moderate, and severe) according to the symptoms, and tried to clarify the distinction between these four levels [17]. Since EDS is a subjective feeling, scoring scales are commonly used, like the Epworth sleepiness scale (ESS) [18]. We used the Chinese Epworth sleepiness scale (CESS), which was validated, with permission from the original author, and published in Quality of Life Research Year 2002 [19].

The strength of association with EDS decreases with increasing age; EDS is more prevalent in the young (<30 years) suggesting the presence of unmet sleep needs and depression [20]. The association of depression with EDS is also stronger in the young [20]. Among the general population, depression is the most significant risk factor for EDS, followed by body mass index, typical sleep duration, smoking, and sleep apnea [20]. A 2018 research reported that high school seniors with EDS had an increased risk for depression. The study also found that sleep deprivation was common among these students [21]. One cross-sectional analysis examined the association between self-reported sleep duration and EDS with indices of obesity, taking into account potential interactions with lifestyle risk factors, such as physical activities [22].

Sleep-deprived behaviors, including watching television, using computers, and having an irregular sleep schedule [23], were another common cause for EDS [24,25,26,27,28]. The association of screen time exposure with sleep and psychological health, including insomnia [29], sleep quality [28], depression, and anxiety [30], have been widely discussed. Students who spend more time using the Internet have less sleeping time and feel higher levels of tiredness [31]. In a South Korean study with a total of 2336 high school students, the results showed that Internet addiction was strongly associated with EDS in adolescents [32]. College students tended to delay their sleep time, which made for short sleep time and high prevalence of sleep problems such as poor sleep quality and insufficient sleep [23]. Computer use has been shown to be associated with poor sleep in previous Taiwan studies [33].

Even though the effects of depression or Internet overuse on EDS among adolescents has been studied in recent years [20,21,32,33], studies on the relationship between screen time exposure, depression, and EDS among college students have been rarely performed. The objectives of this study were (1) to investigate the prevalence of EDS in Taiwanese college students; and (2) to examine the correlation between risk factors (such as depression, screen time exposure, and sleep duration) and EDS in college students. Although we collected data from only two colleges, we believe this is a pioneer study of this topic in Taiwan.

## 2. Materials and Methods

### 2.1. Participants, Study Design and Sampling Method

After taking the national college entrance examination, students in Taiwan were matched to their colleges almost immediately after based on their scores on the examination. This investigation was conducted three months after the examination period so that the students were not affected by this additional stressful factor. The freshmen students of these colleges were mostly from Taipei city and New Taipei city and lived at home.

A cross-sectional study was conducted from July to September 2014. All data were collected from freshmen at two colleges in northern Taiwan. A sample of 3000 freshmen students were randomly selected from the list of all students registered in the colleges, which was obtained from the Academic Affairs Department of these colleges. A total of 2677 students (89.2%) agreed to participate, while students with chronic diseases were excluded. 2656 questionnaires (from 1028 males and 1628 females) were collected. A further thirteen students were excluded because they were taking insomnia medications, leaving a final total of 2643 students (1024 males and 1619 females, mean age of 18.8 ± 1.2 years) with a completion rate of 98.7%.

All students were administered a questionnaire after informed consent was obtained. The standard self-administered questionnaire contained four sections.

### 2.2. Questionnaires

#### 2.2.1. Chinese Epworth Sleepiness Scale (CESS)

Each subject completed the CESS, which is a translated version of a self-rated scale developed in 1991 by Murray Johns [18], and published in Quality of Life Research (2002). The CESS had fairly good retest reliability, correlation co-efficiency from 0.22 to 0.86, and Cronbach’s alpha value of 0.81 [19]. The scale included eight scenarios: sitting and reading; watching television; sitting inactive in a public place (e.g., a theater or a meeting); being a passenger in a car for an hour without a break; lying down to rest in the afternoon when circumstances permit; sitting and talking to someone; sitting quietly after a lunch without alcohol; and in a car while stopped for a few minutes in traffic, to assess the extent and frequency of sleepiness. A rating of 0 (never sleep) to 3 (occasionally sleep) was given to each scenario for the likelihood to sleep.

#### 2.2.2. Brief Symptom Rating Scale (BSRS-5)

Depression was screened using the BSRS-5 [34]. This self-rated questionnaire required respondents to answer whether they have felt tense, blue, irritated, inferior, or had trouble falling asleep in the past week. Responses were rated on a scale of 0 to 4, with 0 being “not at all” and 4 being “extremely”. The BSRS-5 demonstrated good reliability and validity [35,36]. Total scores ranged from 0 to 20, and were divided into four groups: “no symptoms” (0–5), “mild” (6–9), “moderate” (10–14) and “severe” (over 15). The Cronbach’s alpha for BSRS-5 was 0.86, indicating strong internal consistency [37].

#### 2.2.3. Insomnia Symptoms

The Athens Insomnia Scale (AIS), designed to assess the severity of insomnia, uses diagnostic criteria from the International Classification of Diseases (ICD-10) with eight items evaluating sleep onset, night and early-morning waking, sleep time, sleep quality, frequency and duration of complaints, distress caused by the experience of insomnia, and interference with daily functioning [38] The Chinese version of the Athens Insomnia Scale (CAIS) uses items 1–5 of the AIS (night-time symptoms) and scoring from 0 to 3 represent with 0 being “asymptomatic“ to 3 being “severe symptoms”. to for screening and diagnosing insomnia in clinical practice and has satisfactory reliability and validity. The Cronbach’s alpha of internal consistency for these assessments reached 0.82–0.84, and the correlation coefficients of test–retest reliability were 0.84–0.86. The correlation coefficients between the CAIS and Insomnia Self-Assessment Inventory were 0.72–0.76. The suggested CAIS-5 cut-off point for insomnia in the ethnic Chinese population was 5 (area under the curve = 0.90, *p* < 0.01) [39].

### 2.3. Sedentary Behaviors and Physical Activity

#### 2.3.1. Time Spent on Watching Television, Using Computer, and Sleep

Based on previous literature review, leisure activities chosen by people on holidays as compared with those chosen on non-holidays were usually different, regardless of one’s occupation [40]. With this in mind, the mean time students spent watching television, using a computer, and sleeping per day on holidays and non-holidays were recorded separately in our study.

Students tended to watch television after school, mostly at night time. According to previous studies, the total time of computer use, including writing homework and leisure use, was the main factor causing EDS [9,29].

#### 2.3.2. Physical Activity

Students were asked “How frequently do you usually exercise after school/during holiday (weekend day)?”. Four response options (0, 1, 2–3 and ≥4 times/week) were provided. Exercise referred to moderate or vigorous activities, such as jogging, swimming, biking, aerobic dance, and playing ball [40]. All of the measures used in the study were reliable and valid. The Cronbach’s alpha for the sedentary behaviors and physical activity were 0.74, and 0.85 respectively, indicating moderate to strong internal consistency.

### 2.4. Other Factors

Body mass index (BMI) was calculated from the height and weight of each student. Smoking and alcohol drinking habits during the past half-year were also assessed. Participants were also asked to rate their health condition on a scale of 0 to 100 with 0 being “bad heath status” and 100 being “good health status”.

### 2.5. Statistical Analysis

Data analysis were conducted using the software SAS 9.0 (SAS Institute, Inc., Cary, NC, USA) and SPSS 17.0 (SPSS Inc., Chicago, IL, USA). Descriptive analysis was presented as means (with standard deviation, SD) and percentages (%). Student’s *t*-test was used to analyze continuous variables (BMI, BSRS score, heath score, and time spent for sedentary activities); the chi-squared test was used for categorical variables (personal habit and depression groups). Multivariate logistic regressions and multiple linear regressions were used to investigate the association of EDS and different factors among male and female college freshmen.

In multivariable logistic regression, we used the forward selection procedure. The dependent categorical factor in the EDS was dependent on score; a CESS score ≥ 10 was considered with-EDS group, <10 was considered without-EDS group; independent factors included continuous variables (time spent for sedentary activities) and categorical variables (smoking, alcohol drinking, and depression groups).

In multiple linear regression, the dependent factor was the CESS score, and the independent factors were continuous variables, including the BSRS score and time spent in sedentary activities. *P* values of <0.01 and <0.05 were considered to be statistically significant.

### 2.6. Ethical Certification

The study was certificated by the Ethics Committee of Mackay Memorial Hospital (No. 09MMHISO21), and all students were notified of our plan and purpose before our visit.

## 3. Results

A total of 2643 students, 1024 males (38.7%) and 1619 females (61.3%), were assessed. The mean age of the male students was 19.2 (SD: 0.4) years, and of the female students was 18.5 (SD: 0.8) years; less than 10 students were age 20 years or older. There was no statistical difference between male and female students in age (*p* = 0.66). By defining EDS as having CESS scores over 9, the prevalence was found to be 27.1% (male 28.3%; female 26.4%). No statistical differences were seen in mean BMI between the with-EDS and without-EDS groups in both genders. A higher proportion of male students with EDS had smoking and drinking habits. Among those with a depression score over 6, a higher percentage of EDS was noted in each sub-group (mild, moderate, severe depression) comparing to the no depression group. In terms of self-assessments of one’s health status, female students with EDS were more likely to have a lower self-rated health score than male students (*p* < 0.01). Furthermore, female students with EDS were found to spend more time on the computer during non-holidays (*p* = 0.05). Female students with EDS spent significantly less time sleeping than females without EDS on holidays (*p* = 0.02) and non-holidays (*p* = 0.03). Similar patterns were seen in male students although the data were not statistically significant. The characteristics of students are summarized in Table 1. No statistically significant difference was found for insomnia condition and physical activity between those with and without EDS in different genders.

Relevant variables that were selected through the multivariate logistic regression (forward selection procedure) were included as independent variables, such as age, BMI smoking, alcohol, depression, sedentary activities time, and health score (Table 2). Male students with a smoking habit had higher risk of EDS with an odds ratio (OR) of 1.44. Depression increased risk of EDS regardless of gender. The result showed the risk of EDS was elevated with increase in severity of depression: female OR = 2.8/3.71/5.01, and male OR = 3.29/5.07/5.07 for mild/moderate/severe depression, respectively (*p* < 0.05; marginally higher in male severe depression, *p* = 0.08). Female students who spent more time on the computer during non-holidays had higher EDS risk (OR = 1.06, *p* = 0.04), but no statistical significance was found in males.

In Table 3 we sought to score the CESS using different risk factors, which can be useful in clinical screening for severity of EDS, and the analysis was done by multiple linear regression. The result showed that the basic CESS score was 4.97 among all college students; if BSRS score increased by 1 point, CESS score increased by 0.35 point (R^2^: 0.117, *p* = 0.01); if time spent on the computer during non-holidays increased by 1 h, CESS score increased by 0.1 point (R^2^: 0.121, *p* = 0.01); for those whose sleep increased by 1 h during non-holidays, CESS score decreased by 0.1 point (R^2^: 0.122), *p* = 0.02). Among male college freshmen, the results showed that the basic CESS score was 4.67; if BSRS score increased by 1 point, CESS score increased by 0.38 point (R^2^: 0.123, *p* < 0.01). Among female college freshmen, the result showed that the basic CESS score was 4.20; if BSRS score increased by 1 point, CESS score increased by 0.34 point (R^2^: 0.115, *p* < 0.01); if time spent on the computer during non-holidays increased by 1 h, CESS score increased by 0.1 (R^2^: 0.12, *p* = 0.006).

## 4. Discussion

According to our survey, EDS prevalence was 27.1% (male: 28.3%; female: 26.4%), which was higher than that of other general populations (3–13%) [11,12,13,14] and that among high school seniors (17.9%) [16]. The higher EDS prevalence in our study may be explained by the fact that our study mainly targeted college freshmen with a mean age of 18.8 years, a different age group compared to previous studies (children or teenagers) [14,15]. BMI has been mentioned in the literature as an independent factor for EDS [20]. In our study, BMI did not influence EDS, since most of the college freshmen had BMI in the normal range. However, our result was likely to be affected by the younger population chosen for our study, as opposed to older populations used in previous studies [41]. Moderately active and less insomnia were found in both genders with/without EDS. Female students with EDS were more likely to have a lower self-rated health score than male students.

We found that EDS risk was higher in male students with a smoking habit (OR of 1.44). Previous literature reported sleep disturbances can be caused by a nicotine effect in smokers, thereby affecting sleep quality and resulting in EDS. However, we cannot rule out that smoking habits may be related to emotional factors such as depression, which we assessed as another factor [2].

Symptoms of psychological disorders (especially depression) have been linked to disturbances in brain neurotransmitters, thus interrupting nighttime sleep quality and increasing the risk of EDS [21,26]. In our study, we found that depression in both genders was related to increased EDS risk (male OR = 3.7; female OR = 3.0). Increasing in severity of depression was associated with higher OR values (female OR = 2.80/3.71/5.01, male OR = 3.29/5.07/5.07, for mild/moderate/severe depression, respectively, *p* < 0.05; marginally higher in male severe depression, *p* = 0.08). This finding is consistent with that of a previous study in which high school seniors were three times more likely to have strong depression symptoms (OR = 3.04) if they had excessive daytime sleepiness [21].

The prevalence of EDS differed according to gender (25.3% in females, 19.0% in males, *p*-value = 0.036) among respondents with shorter hours of sleep per night. EDS was strongly related to female gender in previous study [42,43]. In our study, the length of sleep was found to have a significant difference between with-EDS and without-EDS groups in the female gender; shorter duration was found in with-EDS group on both holidays (with EDS/without EDS: 9.13/9.14 h, *p* = 0.02) and non-holidays (with EDS/without EDS: 6.90/6.92 h, *p* = 0.03). Sleep duration was not significantly different between the two groups in males (with EDS/without EDS = 8.61/8.73 h on holidays, with EDS/without EDS = 7.00/7.07 h on non-holidays). The previous study also found that sleep deprivation was common among high school seniors. Students reported a mean total sleep time on school nights of only 6.1 h and an increased sleep time of 8.2 h on weekend nights. The American Academy of Sleep Medicine reported that high school students need a little more than nine hours of nightly sleep to maintain sufficient alertness during the day [21]. Our study found that for students whose sleep increased by 1 h during non-holidays, CESS score decreased by 0.1 point (R^2^: 0.1) (*p* < 0.01), which supports the idea that sleep deprivation increases the risk of EDS.

Studies have regarded sleep-time habits as an important factor of EDS [23]. With recent advanced development in computer and internet technology, insufficient sleep due to sleep-deprived habits, such as watching television, playing computer games and using the internet, is becoming more common [31]. In our study, we found that if time spent on the computer during non-holidays increased by 1 h, CESS score increased by 0.1 point (R^2^: 0.121) among all surveyed college freshmen. Female students who spent more time on the computer during non-holidays had higher EDS risk (OR = 1.06, *p* = 0.04), but no statistical significance was found in males. According to one South Korean study of 2336 high school students, the odds of EDS were 5.2-fold greater (95% CI: 2.7–10.2) in Internet addicts and 1.9-fold greater (95%CI: 1.4–2.6) in possible Internet addicts compared to non-addicts [32]. The results of one Taiwan study also showed the negative impact of computer-use among college students. [33].

Previous studies have showed that physical exercise can modulate these two factors, included decreased sympathetic activity and/or an decreased nocturnal hypoxemia, in addition to improving the inflammatory profile of individuals with mild chronic inflammation such as obese individuals [44]. Alves et al. suggested that physical exercise altered cytokine quantity and profile and reduced the effects of cytokines on the central nervous system and more directly on sleep [45]. In our study, moderate physical activity was found in both gender with/without EDS, which was not statistically significant among our subjects. The main effect of the moderate activity level and normal BMI (not obesity) decreased chronic inflammation that affected the sleep quality.

Excessive daytime sleepiness can directly affect the behavior and job performance of the individuals [5,6,7,8,9]. Several causes may be behind excessive daytime sleepiness. Researchers have found that university students who had lower nocturnal sleep duration or an irregular sleep–wake schedule were more likely to report daytime sleepiness. In our study, we did not find severe sleep problems (such as insomnia situation) to be common among our subjects, and this may be because we focused on freshmen who just finished the national college entrance examination. This investigation was conducted three months after the examination period so that the students were not affected by this additional stressful factor. There are some limitations to this study. Our research was a cross-sectional study, which was limited in terms of drawing cause–effect conclusions. Our study focused on colleges in Northern Taiwan but not those of Central or Southern Taiwan, which might result in cultural bias due to different lifestyle habits among the regions. Moreover, each student was asked to recall the length of sleep time without actual objective measurements, which could result in over- or under-estimations. The sleep quality tests were also not further analyzed in detail. However, our research remained one of the few studies investigating EDS among college students, and reported the correlation between depression and time spent on sedentary activities and EDS. The result from this study should be of value to future research studies in this field. Intervention studies, larger multicenter and longitudinal studies, are needed to further explore the causes of EDS and preventive measures to improve EDS in college students.

## 5. Conclusions

Daytime sleepiness was highly prevalent among the college freshmen in this study. EDS significantly predicted depression among college freshmen. Using computers for a long time and less sleep duration during non-holidays increased EDS of college freshmen. Sleep-deprived behaviors and the association between EDS and psychological distress are areas of student well-being that deserve more attention. This study will assist in pointing out important factors causing EDS in the youth population, thus allowing physicians to offer appropriate screening and treatment to these patients.

Besides consideration of potentially sleep-affecting diseases, medications, and sleep hygiene habits in patients with EDS, one should also keep in mind other possibilities, such as sleep duration and quality, depression symptoms, and time spent in front of a screen, in order to provide appropriate management and preventive strategies.

## Figures and Tables

**Table 1 ijerph-16-03148-t001:** Characteristics of the study sample.

Variables	Total (*n* = 2643) Without/With Excessive Daytime Sleepiness (EDS) (*n* = 1926/717, 72.9%/27.1%)					
	Males (*n* = 1024, 38.7%)			Females (*n* = 1619, 61.3%)		
	Without EDS (734, 71.7%)	With EDS (*n* = 290, 28.3%)	*p* Value	Without EDS (*n* = 1192, 72.9%)	With EDS (*n* = 427, 27.1%)	*p* Value
**BMI (kg.m^2^)**, mean (SD)	23.0 (4.8)	22.8 (4.6)	0.52	21.0 (3.8)	21.1 (4.1)	0.54
**Smoking in recent half-year** (*n*, %)			0.04 *			0.82
Yes	116 (15.8)	61 (21.1)		23 (1.9)	9 (2.1)	
No	618 (84.2)	229 (78.9)		1168 (98.1)	419 (97.9)	
**Alcohol in recent half-year** (*n*, %)			0.02 *			0.39
Yes	83 (11.3)	48 (16.6)		42 (3.5)	19 (4.5)	
No	651 (88.7)	242 (83.4)		1149 (96.5)	409 (95.5)	
**Insomnia** (*n*, %)			0.32			0.28
Yes	57 (7.7)	26 (8.8)		127 (10.7)	39 (9.2)	
No	677(92.3)	264 (91.2)		1065 (89.3)	388 (90.8)	
**Depression score** (*n*, %)			<0.01 ^†^			<0.01 ^†^
0–5 (normal)	638 (86.9)	191 (65.9)		984 (82.5)	262 (61.3)	
6–9 (mild)	76 (10.4)	71 (24.5)		160 (13.4)	114 (26.8)	
10–14 (moderate)	18 (2.5)	24 (8.3)		45 (3.8)	47 (11.0)	
≥15 (severe)	2 (0.3)	4 (1.4)		3 (0.3)	4 (0.9)	
**Health score**, mean (SD)	75.2 (13.8)	73.1 (12.6)	0.95	73.4 (12.8)	69.9 (13.9)	<0.01 ^†^
**Exercise frequency (time/week during last three months)** (*n*, %)			0.21			0.26
0	48 (6.6)	35 (11.6)		106 (8.9)	44 (10.4)	
1	190 (25.8)	60 (20.9)		366 (30.7)	149 (34.8)	
2~3	344 (46.9)	148 (51.2)		540 (45.3)	182 (42.7)	
≥4	152 (20.6)	47 (16.3)		180 (15.1)	52 (12.1)	
**Sedentary activities** (hours, mean (SD))						
Watching TV						
Non-holiday	2.37 (1.7)	2.42 (1.7)	0.70	2.70 (1.7)	2.71 (1.7)	0.86
Holiday	3.49 (2.3)	3.43 (2.3)	0.73	4.40 (2.5)	4.44 (2.5)	0.79
Computer (using internet, playing games)						
Non-holiday	3.16 (2.0)	3.33 (2.4)	0.29	2.58 (1.9)	2.83 (2.3)	0.05 *
Holiday	4.70 (2.8)	4.89 (3.0)	0.34	4.08 (2.6)	4.27 (2.6)	0.20
Sleep duration						
Non-holiday	7.07 (1.3)	7.00 (1.4)	0.43	6.92 (1.3)	6.90 (1.3)	0.03 *
Holiday	8.73 (1.9)	8.61 (1.7)	0.32	9.14 (1.6)	9.13 (1.7)	0.02 *

Note: Chi-squared test; *t*-test, independent by groups. * *p* < 0.05, ^†^
*p* < 0.01.

**Table 2 ijerph-16-03148-t002:** Odds ratios of individual factors affecting EDS among male and female students by multivariate logistic regression.

Variables	Males (*n* = 1024, 38.7%)			Females (*n* = 1619, 61.3%)		
	AOR	95% CI	*p* Value	AOR	95% CI	*p* Value
**Personal Habits**						
Smoking (ref: no)						
Yes	1.44	1.01–2.06	0.04 *	1.07	0.47–2.44	0.87
Alcohol (ref: no)						
Yes	1.49	0.99–2.21	0.05 *	1.20	0.67–2.15	0.53
**Depression** (ref: no)						
Mild	3.29	2.26–4.77	<0.01 ^†^	2.80	2.11–3.73	<0.01 ^†^
Moderate	5.07	2.67–9.76	<0.01 ^†^	3.71	2.38–5.77	<0.01 ^†^
Severe	5.07	0.84–10.69	0.08	5.01	1.11–9.68	0.04 *
**Sedentary activities** (continuous variable)						
Watching TV						
Non-holiday	0.99	0.91–1.08	0.88	0.99	0.93–1.07	0.94
Holiday	0.98	0.92–1.04	0.51	1.00	0.96–1.05	0.86
Computer (using internet, playing games)						
Non-holiday	1.03	0.96–1.10	0.38	1.06	1.00–1.12	0.04 *
Holiday	1.03	0.98–1.08	0.29	1.03	0.98–1.07	0.23
Sleep duration						
Non-holiday	0.97	0.87–1.08	0.57	0.98	0.90–1.07	0.68
Holiday	0.99	0.91–1.15	0.96	0.99	0.93–1.07	0.98

Note: Forward selection procedure. * *p* < 0.05, ^†^
*p* < 0.01; AOR: adjusted odds ratios (controlled by age, BMI, smoking, alcohol, depression, sedentary activities time and health score).

**Table 3 ijerph-16-03148-t003:** Individual factors affecting EDS score among total, male and female students by multiple linear regression.

Variables	Regression Coefficient	Standard Error	*p* Value	Cumulative R^2^ × 100
**Total**				
Intercept	4.97	0.33		—
Depression score	0.35	0.02	0.01 ^†^	11.7
Computer, internet using, playing game(non-holiday)	0.10	0.03	0.01 ^†^	12.1
Sleep duration (non-holiday)	−0.10	0.05	0.02 *	12.2
**Males**				
Intercept	4.67	0.15		—
Depression score	0.38	0.03	<0.01 ^†^	12.3
**Females**				
Intercept	4.20	0.15		—
Depression score	0.34	0.02	<0.01 ^†^	11.5
Computer, internet using, playinggame(non-holiday)	0.10	0.04	0.006 ^†^	12.0

Note: * *p* < 0.05, ^†^
*p* < 0.01.

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
