# Peer review of "Association of Depression and Excessive Daytime Sleepiness among Sleep-Deprived College Freshmen in Northern Taiwan"

_ijerph, 2019, doi:10.3390/ijerph16173148_

Round 1
Reviewer 1 Report
I have some comments on the study.
Study design
1. The author did not take into account the influence of a number of factors that have a strong impact on the studied indicator, such as the level of physical activity, academic performance (or degree of involvement in the educational process), whether the study participants experienced the impact of academic stress (exams) before or during the survey, place of residence (at home or in the hostel), from which locality a person came to College (city or village). If these factors are unknown, the author should indicate their absence in the limitations section.
2. Page 3 lines 99-103. The author did not ask the participants of the study what time of day they watched TV, worked on the computer, what determines the degree of influence of these factors on the quality of night sleep and as a consequence on daytime sleepiness. The absence of this information significantly reduces the quality of the study.
3. The author did not evaluate the quality of sleep of the study participants, which is also a significant drawback of the study design.
Introduction
1. At the end of the introduction section, the author should formulate working hypotheses of the study.
Methods
1. The author must indicate Cronbach alpha for all the tests that were used in the study.
2. Page 2 lines 78-79. The study was conducted among college freshmen, but the age range is very large, the author should explain this and indicate the average and standard deviation of the indicator.
3. Page 3 lines 104-107. The author should use in the analysis not raw BMI values, but corrected by age and sex.
4. Table 3. The author should indicate in the note that he used in the Table and in the text not raw values of R2, but values of R2 *100%.
Conclusion
In the conclusion section, the author should indicate the main results of the study and note which hypotheses were confirmed and which were not.
Reviewer 2 Report
Firstly, I congratulate to the authors! I consider that they have done a great work! Thank you for giving me the opportunity to revise this manuscript. I found this paper particularly interesting because is focused in an important problem in university students as is the excessive daytime sleepiness.
I appreciate you will have put in a lot of effort in preparing your paper and there are several points which I offer to assist with further developing the manuscript. These points are listed under the subheadings.
Introduction:
In the line 42 Page 1, it appears “CESS”. Can you explain the significance of this acronym?Methods:
Did you do sampling? Did you used a program to calculate sample size? What was the reference population? Please, provide more information about it! How was the data collection? Did you do face-to-face interviews? Did you used an online questionnaire? Please, provide more information about it! Can you provide the reliability and validity of the CESS? How was evaluated the health status or health condition? Did you used a validated scale?Discussion:
I think that it would have been interesting to talk about the relationship between depression and excessive daytime sleepiness with physical exercise.References:
I think that your literature should be the most recent on this topic; there are 3 of 40 references that belong to last five years. Try to include and change by the most up to date international literature!
Finally, the authors present the differing aspects of your research in an appropriate way and this allows the reader to follow the study and understand the approaches, analysis and discussion of findings which are embedded within the aim of to identify the prevalence and risk factors of EDS among college students. I hope that the authors can do the revisions of the manuscript for to can be accept it in International Journal of Environmental Research and Public Health.
Author Response
Please see attachement.

Round 2
Reviewer 1 Report
I have no comments.
Author Response
Dear Reviewer:
Thanks for your comments.
Sincerely
MT Tsou
Reviewer 2 Report
Congratulations! I'm pleased you have amended the paper which is more much more readable and accessible to a wider readership. The authors did a great job in revising the manuscript and thoroughly addressed reviewer comments.
Author Response
Dear reviewer:
Thanks for your comments.
Sincerely
MT Tsou